# Structural Health Monitoring Based on Acoustic Emissions: Validation on a Prestressed Concrete Bridge Tested to Failure

**DOI:** 10.3390/s20247272

**Published:** 2020-12-18

**Authors:** Daniel Tonelli, Michele Luchetta, Francesco Rossi, Placido Migliorino, Daniele Zonta

**Affiliations:** 1Department of Civil, Environmental and Mechanical Engineering, University of Trento, 38123 Trento, Italy; michele.luchetta@unitn.it (M.L.); daniele.zonta@unitn.it (D.Z.); 2Department of Civil and Environmental Engineering, University of Strathclyde, Glasgow G1 1XJ, UK; francesco.rossi@strath.ac.uk; 3Ministry of Infrastructures and Transport, 00157 Rome, Italy; placido.migliorino@mit.gov.it

**Keywords:** acoustic emissions, damage detection, crack initiation, crack propagation, prestressed concrete bridge, load test, structural health monitoring

## Abstract

The increasing number of bridges approaching their design life has prompted researchers and operators to develop innovative structural health monitoring (SHM) techniques. An acoustic emissions (AE) method is a passive SHM approach based on the detection of elastic waves in structural components generated by damages, such as the initiation and propagation of cracks in concrete and the failure of steel wires. In this paper, we discuss the effectiveness of AE techniques by analyzing records acquired during a load test on a full-size prestressed concrete bridge span. The bridge is a 1968 structure currently decommissioned but perfectly representative, by type, age, and deterioration state of similar bridges in operation on the Italian highway network. It underwent a sequence of loading and unloading cycles with a progressively increasing load up to failure. We analyzed the AE signals recorded during the load test and examined how far their features (number of hits, amplitude, signal strength, and peak frequency) allow us to detect, quantify, and classify damages. We conclude that AE can be successfully used in permanent monitoring to provide information on the cracking state and the maximum load withstood. They can also be used as a non-destructive technique to recognize whether a structural member is cracked. Finally, we noticed that AE allow classifying different types of damage, although further experiments are needed to establish and validate a robust classification procedure.

## 1. Introduction

An increasing number of civil infrastructures are approaching or exceeding their initial design life. For instance, the average age of bridges in the USA is estimated to be 46 years [1]; similarly, 70% of highway bridges in Italy were reportedly built around the 1970s [2]. As infrastructure ages, the effort required by operators to identify unsafe structural conditions increases. The benefit of structural health monitoring (SHM) to bridge management has been extensively analyzed in the literature, see for instance [3]; SHM allows the early detection of possible damages resulting from the natural deterioration of structural materials, and to optimize decision over maintenance, repair, and reconstruction of the bridge asset [4,5].

The growing interest in SHM for infrastructure operators and the recent technological progress have encouraged the research community to study and develop innovative sensors and monitoring methods. An incomplete list includes the Ground Penetration Radar (GPR), or Georadar [6,7], the Reflectometric Impulse Measurement Technology (RIMT) [8,9,10], the Time Domain Reflectometry (TDR) [11,12], the Strand-Cutting test [13,14,15], the Core Drilling Method (CDM) [16,17], and the Acoustic Emissions technique (AE) [18]. All these techniques have been studied in laboratory experiments and in-service structure monitoring; however, the absence of standardized procedures and the unavoidable need of experts to interpret results make these technologies unready for an extensive application on civil infrastructure yet. It is necessary to validate such technologies and define specific protocols to guarantee the accuracy and reproducibility of their results, which any qualified practitioner should be able to interpret. That is necessary to obtain effective monitoring information for infrastructure management [19,20].

In this contribution, we focus on the AE technique. It is a passive monitoring approach based on the detection of elastic waves in structural components generated by damages, such as the initiation and propagation of cracks, the failure of steel wires, and the failure of bonds [21]. Its primary goal is to detect, locate, and assess the intensity of damage [18] in a non-invasive way, both when the structure is in-service and during load tests. Its application in SHM started much later compared to other fields, such as the aerospace industry [21]. The interest has increased because elastic waves generated by damages propagate throughout the structure; therefore, it is possible to remotely detect damages in areas that are not easily accessible to visual inspections and direct measurements [21].

Numerous laboratory studies on crack detection based on AE have been conducted on specimens representative of in-service structures, both on steel members [22,23,24] and concrete samples [25,26,27]. In addition, field-testing applications have been carried out on large-scale structures, such as bridges, nuclear power plants, containment structures like silos, bins, and water storage tanks [28], and prestressed concrete pipes exposed to corrosion phenomena [29]. Applications of AE for damage monitoring in masonry structures are also reported in the literature [30].

The use of AE in bridge monitoring dates back to the 1970s when Pollock and Smith monitored for the first time a portable military bridge subjected to proof testing [31]. After this experiment, AE technology has been used in numerous field bridge testing applications [18], addressing the detection of concrete cracks initiation and propagation, the development of fatigue cracks in steel members, the failure of prestressed tendons in prestressed reinforced concrete elements, and the break of wires in cable structures. A comprehensive review of AE monitoring applications on bridges from 1970 to 2010 is found in [18].

Golaski et al. [32] used AE monitoring for the safety assessment of five full-scale bridges of different types: reinforced concrete, prestressed concrete—both post-tensioned and pre-tensioned—and steel-concrete composite. They claimed that the AE method is useful for evaluating the integrity of bridges; however, individual evaluation-criteria must be selected for each structural type. Nair and Cai [18] focused on the condition assessment of a prestressed concrete slab-on-girder bridge and a steel bridge with a concrete deck under live loads. Anay et al. [33] worked on the damage identification in a three-span prestressed concrete girder bridge with pre-existent inclined cracks during a load test and used AE tests to classify crack extensions as stable or unstable.

Many experiments have specifically addressed the crack opening and propagation in concrete elements. Chataigner et al. [34] realized several real-size experimental investigations on a prestressed concrete girder taken from a decommissioned viaduct, carrying out both flexure and shear tests up to failure and using several measurements methods, including AE. They claimed that it is possible to learn additional information on the structure’s state of damage by comparing results from traditional crack-opening sensors and AE sensors. They also compared the results with the prediction from a finite-element-model and with the results of an autopsy of the beam carried out by hydro demolition; they found a good correlation, even though they observed undetected damages during the autopsy. Recently, Ma and Du [35] applied a machine learning algorithm based on a Deep Neural Network model to combine some of the most common AE signal parameters to assess the rate of crack-opening on prestressed concrete structures. They found interesting results on the correlation between AE parameters and crack-opening events for different loads; however, they recognized some limitations in the approach.

Other research works have focused on AE as a means to detect failures of tendons in prestressed concrete bridges. Fricker and Vogel [36] monitored a small prestressed concrete bridge to evaluate the performance of permanent AE monitoring and demonstrated that it is possible to detect wire breaks with good localization accuracy. Yuyama et al. [37] carried out both laboratory and field tests: laboratory tests were conducted on three types of post-tensioned beams (with steel bars, strands, and parallel wire cables), while field tests were performed on two post-tensioned bridges. They found that it is easy to discriminate meaningful AE due to wire breaks from traffic noise and hammering and claimed that AE is a promising method to detect and locate wire breaks. Shiotani et al. [38] studied cable breakages and the subsequent failure process using a full-scale post-tensioned prestressed concrete beam subjected to a four-point bending; they simulated the breakage of cables by reducing their stress in turn and identified breakage areas and failure areas of grout material based on AE.

The AE analysis has recently addressed the problem of corrosion in prestressed concrete structural components exposed to saltwater tides and splashing. Vélez et al. [39] carried out some laboratory experiments on a full-scale prestressed concrete specimen representative of outer portions of bridge piles exposed for one year to saltwater wet/dry-cycles mimicking natural tidal action; they stated that the AE technique is efficient to evaluate and detect corrosive phenomena.

In summary, most of the experimental verifications of AE techniques reported in the literature are based on tests carried out in laboratory conditions on specimens or individual structural elements. There are also a number of works that investigate the performance of AE methods on full bridges in operation. However, not that surprisingly, these experimental works are typically concerned with damage states that do not jeopardize the safety of the bridge, and they do not provide direct verification on how well an AE technique performs when the bridge is close to collapse. Indeed, we are not aware of any experiment on a full-scale bridge that verifies the capacity of AE to provide an early warning when the bridge is approaching its ultimate state.

In this paper, we wish to bridge this gap by analyzing and discussing the AE recorded during a load test on a full-size prestressed concrete bridge span, carried out up to the bridge failure. The bridge, the Alveo Vecchio viaduct, is a 1968 structure currently decommissioned but perfectly representative, by type, age, and deterioration state of similar bridges currently in operation on the Italian highway network. Our goal is to discuss how the AE change while the load progressively increases and compare the results with those provided by other sensors installed on the structure: linear variable differential transformers (LVDT) for the cracks detection and rotary variable differential transformers (RVDT) for the bridge deflection. We aim to verify whether an AE monitoring system can provide useful information to (i) discriminate whether the viaduct has pre-existent damages, such as concrete cracks or broken steel wires; (ii) identify the opening of the first crack; (iii) single out the maximum load withstood by the viaduct; and (iv) recognize different types of damages. Particularly, we wish to identify the most sensitive features of AE that deserve to be extracted to establish the level, extent, and type of damage on a prestressed concrete bridge.

For those readers who are not familiar with AE, in the following Section 2, we briefly summarize the physical principle and the technology at the basis of the method, along with the most typical signal analysis techniques. Then, in Section 3, we describe the Alveo Vecchio case study, including details of the monitoring system installed and of the load test carried out. Section 4 reports the results of the AE acquired during the different phases of the test. We discuss these results in Section 5, by comparing them with the evidence recorded by LVDT and RVDT sensors. Finally, in Section 6, we draw conclusions about the application of the AE technique to real-life prestressed concrete bridges.

## 2. AE Principle and Observable Quantities

### 2.1. Phenomenon and Technology

The AE is a phenomenon in which transient elastic waves are generated by the rapid release of strain energy from a localized source due to microstructural changes in the material [40]. Elastic waves travel into the material and move to the surface of a structural element where sensors can detect them. Therefore, an AE monitoring system requires two components: a source, such as a crack propagation or a tendon failure; and a transducer, which receives and acquires the elastic wave [18]. Figure 1 shows the working principle of an AE monitoring system.

An elastic wave is a combination of longitudinal, transverse, and reflected waves, with a broadband frequency range from kHz to MHz [21]. Even though they are called acoustic emissions, elastic waves are neither acoustic (from 2 kHz to 20 kHz) nor ultrasonic (over 20 kHz) [41]. AE sensors are typically piezoelectric or PZT devices that transform the motion produced by the transient elastic wave into an electrical signal, which is digitized and stored [42]. The selection of the transducer’s sensitivity and frequency response is critical for the effectiveness of the AE technique and depends on the characteristics of the monitored structure [43]. Capacitive MEMS AE transducers have been recently designed and tested [44,45]; they are smaller and less expensive than piezometric sensors but have limited sensitivity and a working direction only normal to the surface on which they are installed [21].

When an elastic wave reaches the sensor, it is transduced into an electrical signal, recorded, amplified, and typically represented in a diagram with the time expressed in seconds (s) on the horizontal axis and the signal amplitude expressed in volts (V) on the vertical axis. The signal is usually affected by background and environmental noise due to the wind and passers-by; therefore, the reduction of such noises requires a band-pass filter [41].

### 2.2. AE Signal Parameters

The electrical signal identifies an acoustic event, also called a hit [46], when it crosses a certain threshold, expressed in volts (V) or similarly in decibels (dB). This threshold is defined as the minimum amplitude that the signal must have to be considered in the analysis [47]: typical values for reinforced concrete structures are around 40–45 dB [48], but sometimes it can be up to 60 dB [37,49]. Moreover, the signal must cross the threshold at least three times consecutively to be one hit.

A hit can be described by characteristic parameters [47], which are defined in the time-domain, as represented in Figure 2, or in the frequency-domain, as represented in Figure 3. Here is a summary of the parameters we considered in our analysis.

Amplitude: it is the maximum amplitude of the signal in the time-domain after its amplification. It is expressed in decibels and V_ref_ = 1 μV from the sensor corresponds to 0 dB.Duration: it is the time interval between the first and the last threshold-crossing of a hit.Count: it is the number of times that the signal exceeds the threshold within the duration: it strongly depends on the threshold and the sampling frequency.Signal strength (energy): it is the measured area of the rectified signal envelope (MARSE). Typically, it includes the absolute value of areas of both the positive and negative envelopes. Its unit of measure is Volts × second [V·s], and it is a function of both the amplitude and the duration. It is preferred over count to interpret the magnitude of the event.Peak frequency: it is the frequency corresponding to the peak observed in the power spectrum resulting from an FFT (Fast Fourier Transformation) of the signal.

To discriminate different acoustic events, we must select three time-parameters, PDT, HDT, and HLT, [50,51,52], represented in Figure 4. Their choice is critical for the correct identification of hits.

Peak definition time (PDT): it is the time after the peak amplitude in which a new greater peak amplitude can replace the original one; after the PDT has expired, the original peak-amplitude is not replaced.Hit definition time (HDT): it is the time after the last threshold-crossing that defines the end of the hit.Hit lockout time (HLT): it is the time after the HDT during which a threshold-crossing will not trig a new hit. A new hit can start only after the HLT has expired.

### 2.3. AE Analysis for Load Tests

A structural element subjected to loading and unloading cycles experiences a propagation of damages and emits acoustic waves only when the previous maximum load level is exceeded [18]. The absence of AE during a loading phase is called the Kaiser effect [53] and happens only with an elastic behavior of the material. In the case of plastic deformations, the Kaiser effect is violated, and acoustic waves are emitted during all the loading phase; this phenomenon is called the Felicity Effect [53]. The Kaiser and the Felicity effects identified during load tests can highlight the presence of flaws or other structural damages and help to assess the integrity of the structural element.

## 3. Case Study of a Prestressed Concrete Bridge Tested to Failure

### 3.1. Alveo Vecchio Viaduct

The Alveo Vecchio viaduct is part of the old track of the A16 Napoli-Canosa Italian highway. It was built in 1968 and decommissioned in 2005, after a landslide hit and displaced one of its piers. It is representative of 70% of the Italian highway bridges in terms of the structural scheme, construction technology, maintenance state, age, and deterioration [2]. Figure 5 shows a top view, a lateral view, and a cross-section of the viaduct.

The viaduct consists of two structurally independent decks, one for each carriageway, each of them made of three 32.5 m long prestressed concrete simply-supported spans. Each span consists of four prestressed concrete girders of depth 2 m, which support a 20 cm thick concrete deck slab. The prestressing was applied through 14 post-tensioned cables per each girder, with an initial jacking tension of 1250 MPa. Each cable has an ultimate strength of 1700 MPa and a yielding strength of 1450 MPa. The wall piers are 3.30 m high and have deep foundations consisting of eight piles 23 m long with a diameter of 1.2 m. Additional information about the structure is found in [54]. Table 1 reports the mechanical properties of materials resulting from an extensive sampling and testing campaign performed according to the Italian standard on SHM [55]. In 2005, a landslide hit the C1sx and C1dx spans (see the top view of Figure 5). It resulted in the collapse of span C1dx and the rotation and translation of Pier 1. In 2019, an inspection and a preliminary load test of the piers and abutments reported that the landslide did not affect spans C3sx and C3dx. Therefore, we chose span C3sx for the load test reported in this paper.

### 3.2. Structural Health Monitoring System

The monitoring system designed for the load test consists of 119 sensors divided into eight types: wire displacement sensors (RVDT), deformation sensors (strain gauges), crack-opening sensors (LVDT), electronic level, temperature sensors (RTD), inclinometers, accelerometers, and AE sensors. Furthermore, we monitored air temperature, air humidity, and wind speed. Details about the SHM system installed on the viaduct are in [56] (in Italian).

We focus on wire displacement sensors, crack-opening sensors, and AE sensors. Figure 6 shows the layout of these sensors on the viaduct, while Table 2 reports their technical features. The LVDT sensors are Gefran PZ12, while the RVDT sensors are PT1DC from Celesco Transducer Products, Inc. Their acquisition system consists of a central acquisition unit with eight signal conditioning units both from IO Tech Group Ltd. They measure displacements with a sampling frequency of 1 kHz and record data continuously. On the other hand, the AE sensors are Endevco^®^ Isotron^®^ wide frequency bandwidth accelerometer, Model 42A18. Their acquisition system is the WaveBook/516E from IO Tech Group Ltd. They measure accelerations with a sampling frequency of 10 kHz and record a 2100 ms long sample every time the acceleration exceeds 10 mg (time t_0_ of the sample). The sample starts 100 ms before t_0_ and ends 2000 ms after that.

### 3.3. Load-Test Protocol

The load-test protocol consists of five loading and unloading phases with a progressive number of steel ballast weights with a size of 2.35 × 1.84 × 0.45 m and a weight of 100 kN each. They are placed in the middle of the span C3sx in layers of 12 ballast each. The load unit is 2400 kN: two layers of ballast, which produces a bending moment in the girders’ middle cross-section of 4200 kNm, corresponding to the load effect resulting from the design traffic load [54]. Figure 7 illustrates the five loading phases, identified with codes P1 (1200 kN), P2 (2400 kN), P3 (4800 kN), P4 (7200 kN), and P5 (bridge’s ultimate capacity). Figure 8 shows a picture of the bridge in phase 5, loaded with 93 ballast—9300 kN. Details about the load-test protocol are in [56] (in Italian).

## 4. Results of the Case Study

In this section, we report the most important results about the bridge deflection and the crack opening at the bottom of the prestressed girders. Data come from the sensors installed on the external girder T1, which experienced the greatest deflection and deformation. Figure 6 represents the locations of girders, displacement sensors (RVDT), and crack-opening sensors (LVDT).

Then, we report some significant results from the two AE sensors, T2AE1 and T2EA2, installed on the girder T2 during the loading phases P3 and P4. No AE sensors have been installed on the girder T1, and no relevant results came from the last phase P5 due to the damage of such sensors during that phase. The location of AE sensors is also represented in Figure 6. We analyze raw-data samples with the software MATLAB^®^. We did not use any other commercial software. The parameters investigated include amplitude, signal strength, and peak frequency. Table 3 shows the values of the threshold, the sampling frequency, the high-pass filter, and the three time-parameters PDT, HDT, and HLT, which we calibrated to recognize correctly at least 95% of the hits from a sample of 500 AE randomly extracted from those recorded in the loading phases P3 and P4. We set the threshold on 60 dB to reject all the signals identified as noise in the first part of the loading phase P3; this value is consistent with other studies in the literature [37,49].

During the four days of load test, the temperature was between 18 °C and 34 °C; the relative humidity was in the range 56–60%.

### 4.1. Results from Displacement and Crack-Opening Transducers

Figure 9a shows the vertical displacements of the girder T1 recorded by RVDT sensors during the different phases of the load test, while Figure 9b shows the load-deflection curve of the girder T1 midspan, along with its envelope representing its trilinear idealized flexural response. It is easy to recognize stage I—elastic, stage II—cracks initiation and propagation, and stage III—after the yielding of post-tensioned cables. According to this trilateral load-deflection model of the girder T1, the first-crack load is 3700 kN, while the ultimate capacity is 8700 kN.

Regarding the cracks opening, Figure 10 shows the longitudinal strain recorded by LVDT sensors at the bottom of the middle cross-section of girder T1: Figure 10a is about the loading phase P3—4800 kN, while Figure 10b is about the loading phase P4—7200 kN. In Figure 10a, a change from stage I—elastic—and state II—cracked is visible, corresponding to a load of 3300 kN. In contrast, Figure 10b shows a softer change in the behavior, confirming the cracked condition of the girder from the beginning of P4. The colors green, yellow, and red in the background of graphs represent the layers of steel ballast weights loaded on the viaduct: two green layers for 2400 kN, two additional yellow layers for 4800 kN, and two additional red layers for 7200 kN (see Figure 7 for the load-test protocol).

Figure 11 is a picture of the visible cracks (on girder T1) opened during the loading phase P3. Figure 12 is a picture of the visible cracks (on girder T1) opened during the loading phase P4.

Additional results from the monitoring system, like girders’ deflections, strain measurements, rotations of pier and abutment, and temperatures, are in the load test report [57] (in Italian).

### 4.2. Results from AE Sensors—P3 4800 kN

In this subsection, we report the results of the analysis performed on the AE acquired during the loading phase P3 4800 kN. In Figure 13, we report four graphs with the load from 0 to 4800 kN on the horizontal axis and the amplitude (Figure 13a), the cumulative number of hits (Figure 13b), the signal strength—MARSE (Figure 13c), and the cumulative MARSE (Figure 13d) on the vertical axis. In all of them, the red dashed line represents the first-crack load: 3300 KN; it has been identified by the crack-opening sensors and represented also in Figure 10a. The colors green, yellow, and red in the background of graphs represent the layers of loads on the viaduct, as explained in Section 4.1.

To discriminate different types of damage, such as concrete cracks and failure of steel wires, we analyzed the AE in the frequency domain. We aimed to investigate the presence of clusters in amplitude—peak-frequency graphs and load—peak-frequency graphs. Such clusters may represent different sources of the elastic waves [47]. Figure 14a shows these graphs for both the sensors installed on the girder T2. Two clusters are clearly visible in data from sensor T2AE1, while data are more scattered from sensor T2AE2. We reported the distributions of peak-frequencies from AE recorded by both sensors in a histogram represented by Figure 14b.

Finally, we compared the results from the AE acquired on the girder T2 with the results from the crack-opening sensors LVDT installed on the girder T1. Figure 15a shows the hit amplitudes recorded during the loading phase P3, Figure 15b shows the cumulative MARSE, and Figure 15c shows the strain at the bottom of the girder T1. The blue dashed line at 4300 kN represents the value of the load when we recorded the AE generated by the opening of the first crack in the girder T2; after that, AE increase considerably. In contrast, the red dashed line represents the first-crack load (3300 kN) identified by LVDT, as already explained in Section 4.1.

### 4.3. Results from AE Sensors—P4 7200 kN

The same graphs as those shown by Figure 13, Figure 14 and Figure 15 for the loading phase P3 are here reported for the loading phase P4 in Figure 16, Figure 17 and Figure 18.

## 5. Discussion of Results

We discuss the results of the AE analysis reported in the previous section. Here, we start with the loading phase P3; then, we move on to the phase P4; finally, we present a comparison between results from phases P3 and P4 to identify what AE can effectively suggest in terms of pre-existent cracks or other damages, maximum load withstood, opening of the first crack, and damage recognition.

### 5.1. Discussion of AE Results from P3 4800 kN

Based on the AE results of the loading phase P3, we can identify the opening of the first crack on the girder T2. Indeed, the absence of cracks for low values of the load suggests that the girder T2 was not cracked before the beginning of P3 [53]. We can recognize the opening of the first crack in the graph of the amplitude plotted against the load (Figure 13a and Figure 15a) as the first AE after which the number of AE increases significantly. We can identify it also in the graph of the cumulative MARSE plotted against the load (Figure 13d and Figure 15b) as the point in which the curve has a sharp change of slope. In particular, the first crack opens for a load of 4300 kN, represented by blue dashed lines in those graphs. Figure 13a also shows some AE for loads lower than 4300 kN. However, they are characterized by low values of the MARSE (Figure 13c), which do not change the slope in the cumulative MARSE (Figure 13d); therefore, they are probably related to bearing deformations or thermal effects [33,58]. On the other hand, the crack-opening sensors (LVDT) at the bottom of the girder T1 suggest that the first crack opens for a load of 3300 kN (Figure 15c). The difference between these two values is mainly due to the difference in the girders monitored by the two technologies: girder T1 by LVDTs, and girder T2 by AE sensors. Indeed, during the load test, the girder T1 experienced the greatest deflection and deformation; therefore, it is reasonable to assume that cracks have opened first on the girder T1 and then on the girder T2. As a result, we can conclude that the AE technique can effectively identify the opening of the first crack. However, additional studies and load tests are necessary to directly compare the observation from AE sensors and crack-opening sensors installed on the same girder.

As far as the analysis in the frequency domain is concerned, Figure 14 shows two clusters in data from the sensor T2AE1 (blue dots): around 1 kHz and 2.5 kHz. They may represent different sources of the elastic waves, such as different types of damage; for instance, some authors have recognized that tensile-cracks have a higher average frequency than shear-cracks’ [59,60]. On the other hand, data from the sensor T2EA2 (magenta dots) are more scattered. The reason for this difference is not clear since the two sensors were placed on opposite sides of the same girder’s middle cross-section. The outcomes of our experiment are not enough to univocally establish which types of damage correspond to those clusters. That would require multiple load tests on bridges monitored by AE sensors carried on until different loads followed by autopsies of the monitored girders; that would allow correlating the damages occurred with the AE acquired in each test. In conclusion, AE can recognize clusters of acoustic events, which are probably related to different types of damage. However, further experiments are needed to establish and validate a robust identification procedure.

### 5.2. Discussion of AE Results from P4 7200 kN

Results from AE acquired during the loading phase P4 confirm that the AE technique can effectively discriminate whether a prestressed concrete structure is affected by pre-existent cracks and identify the maximum load withstood by the structure. Indeed, the amplitude—load graph of data recorded during P4 (Figure 16a and Figure 18a) shows several AE for low values of the load. These AE are generated by the friction between the two surfaces of a crack [61]: this confirms the presence of pre-existent cracks in the girder T2 before the beginning of P4. On the other hand, the cumulative-MARSE—load graph (Figure 16d and Figure 18b) shows a change in the curve slope around 4800 kN: this confirms that the maximum load withstood by this viaduct is 4800 kN (at the end of P3) and that the damage propagation starts only after that the maximum load withstood has been exceeded. This last consideration is in line with the Kaiser and Felicity effects, which we will focus on in Section 5.3. The LVDTs installed on the girder T1 (Figure 18c) confirm that such girder experiences a non-elastic behavior starting from the beginning of the loading phase P4, exceeding a strain of 0.8 µε for a load of 3300 kN, while the strain was around 0.2 µε for 3300 kN during P3. Since AE allow to recognize the maximum load withstood by a bridge, their application is particularly effective in statically indeterminate structures designed to achieve robustness [62], where the ultimate capacity is preceded by a progressive state of damage.

As far as the classification of different types of damage based on the AE technique is concerned, in the MARSE—load graph (Figure 16c), we can recognize three events between 5200 kN and 5800 kN with a signal strength around ten times higher than the others. They may have been generated by different sources than concrete cracks, such as post-tensioned steel wires failure. Again, a campaign of load tests on girders monitored by AE sensors followed by their autopsy would help in the correlation between the AE results and the types of damage that occurred during the load tests. Such a campaign would offer useful insights also into the difference between results from sensors T2AE1 and T2AE2 in the frequency domain (Figure 17): data from the first sensor point out two clusters around 1 kHz and 2.8 kHz, while data from the second sensor provide more scattered peak-frequencies.

### 5.3. Comparison of Results from P3 and P4 for Low Values of the Load (0–2400 kN)

Permanent AE monitoring is supposed to provide information also on the condition state of in-service structures, which typically do not experience a load exceeding the load of the design test. Therefore, AE monitoring is effective only if it allows us to understand whether that structure has pre-existent cracks only based on data acquired during low values of the load. In our case study, the maximum weight allowed to transit on the viaduct produces a bending moment equal to the one produced by the unit load 2400 kN. Therefore, AE monitoring is effective only if it can identify a difference in the viaduct behavior in the cases without and with pre-existent cracks only based on data acquired between 0 kN and 2400 kN. Consequently, the comparison between the results from the loading phase P3 and P4 in the range of load 0–2400 kN is particularly interesting. Indeed, the results from the phase P3 are those from a structure without pre-existent cracks, whose response is elastic until the first crack occurs at around 3300 kN; in contrast, results from the phase P4 are those from a structure with pre-existent cracks.

Figure 19a,b show the amplitude–load graphs of AE acquired during loading phases P3 and P4, respectively. Their difference is visible even for low values of the load: the girder with pre-existent cracks (T2 during P4) emits a higher number of AE than the one without cracks (T2 during P3). The cumulative number of hits–load graph (Figure 19c) and the cumulative MARSE–load graph (Figure 19d) show this difference even more clearly: solid lines (T2 during P3) are almost linear, while dotted lines (T2 during P4) have a sharp change in the slope between 1200 kN and 2400 kN.

The difference between P3 and P4 in terms of the cumulative number of hits is consistent with the Kaiser and the Felicity effects, which are visible in Figure 20. Note that Figure 20 is the union of the cumulative number of hits during P3 (Figure 13b), during P4 (Figure 16b), and during the unloading phase between P3 and P4. During the loading phases P3, it is possible to observe the Kaiser effect [53] since any permanent damages have never occurred within the girder before P3: as the load increases from 0 kN to 4800 kN, no AE are generated until the exceedance of the first-crack load. In contrast, during the loading phase P4, it is possible to observe the Felicity effect [53] since some permanent damages have occurred in the previous phase: as the load increases from 0 kN to 7200 kN, some AE are generated starting from 1200 kN, before the previous maximum load (4800 kN) is exceeded. This difference in the AE recorded during phases P3 and P4 for low values of the load points out a difference in the structural behavior and can discriminate the presence of pre-existent cracks.

## 6. Conclusions

In this contribution, we presented an application of the AE technique to a real-life case study, the Alveo Vecchio viaduct, a prestressed-concrete highway-viaduct that underwent a sequence of loading and unloading cycles with progressively increasing loads up to three times the design load. Each test phase was monitored by an extensive monitoring system, which included a network of AE sensors. The most significant results have been observed during test phases P3 (maximum load of 4800 kN, corresponding to two times the design load) and P4 (maximum load of 7200 kN, corresponding to three times the design load).

We analyzed the AE signals recorded during the load test and extracted the following parameters: amplitude, signal strength (MARSE), and peak frequency. We examined how far these signal features can detect, quantify, and classify damage on the bridge. The main outcomes of this research are:

(i) AE allow easy recognition of whether a prestressed concrete bridge is cracked or not: a bridge with pre-existent cracks produces several AE under service load, whereas virtually no hit is recorded on a bridge with no cracks. The different behavior is evident by plotting the amplitude, the cumulative number of hits, or the cumulative MARSE against the load (compare Figure 15a,b, and Figure 19a with Figure 18a,b, and Figure 19b, and note the difference between the curves with label DAMAGE and NO DAMAGE in Figure 19c,d).

(ii) AE also allow clear identification of the opening of the first crack, as this is typically accompanied by the first-time emission of a high-intensity signal, followed by several more as the cracks propagate. The opening of the first crack is easily detected by plotting the amplitude or the cumulative number of hits against the load (see Figure 13a,b). The first-crack also corresponds to the first of a series of AE with a high MARSE in the MARSE graph and to a sharp change in the slope in the cumulative MARSE graph (see Figure 13c,d).

(iii) The AE technique also allows us to identify the maximum load withstood by the bridge: it corresponds to a sharp change in the slope of the cumulative MARSE graph, as shown in Figure 18b.

(iv) In principle, it is possible to classify the damage by type by analyzing the AE in the frequency domain. In our experiment, the AE occur grouped in well-separated clusters in the amplitude–peak frequency graph, suggesting that each cluster corresponds to a different source of AE, and therefore to a different damage episode. While it is possible, in principle, with further experiments, to define a general correlation between clusters and types of damage (cracking, deboning, failure of steel wires, etc.), the outcomes of our experiment are not enough to univocally establish such a correlation.

In summary, the outcomes of this experiment suggest that AE can be used successfully in permanent monitoring of prestressed concrete bridges to provide information on the cracking state and maximum load withstand. They can also be used as a non-destructive technique in short-term monitoring to discriminate whether a structural member has pre-existent cracks or not. Apparently, AE also allow the classification of different types of damage, although further experiments are needed to establish and validate a robust identification procedure.

## Figures and Tables

**Figure 1 sensors-20-07272-f001:**
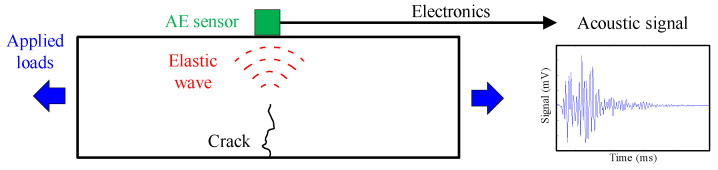
Working principle of an AE monitoring system.

**Figure 2 sensors-20-07272-f002:**
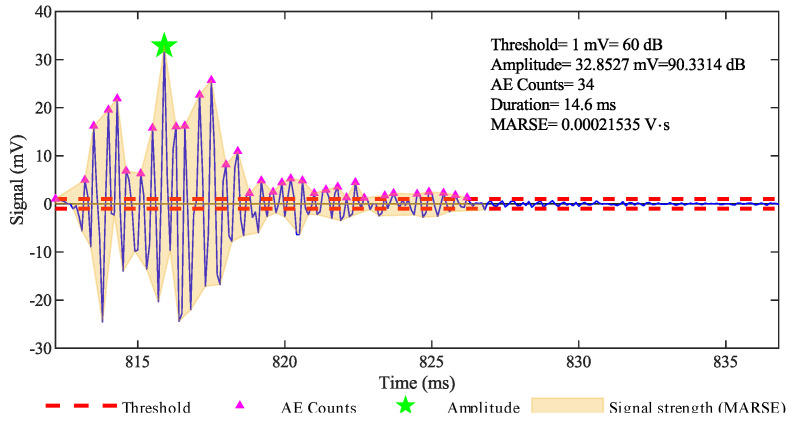
AE signal and parameters expressed in the time-domain.

**Figure 3 sensors-20-07272-f003:**
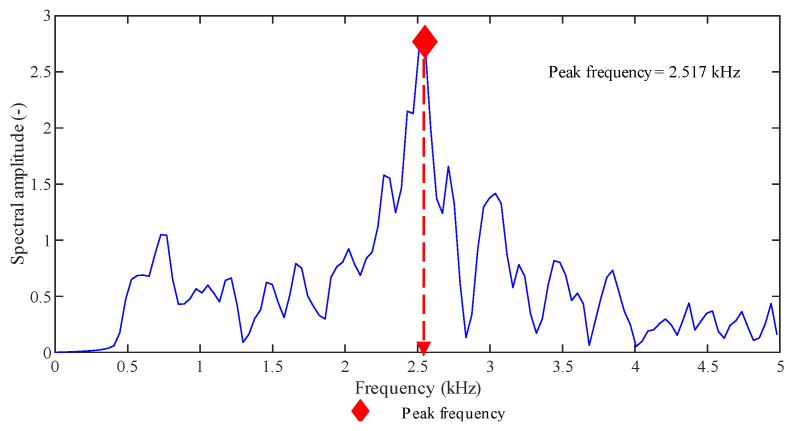
AE signal and parameters expressed in the frequency-domain.

**Figure 4 sensors-20-07272-f004:**
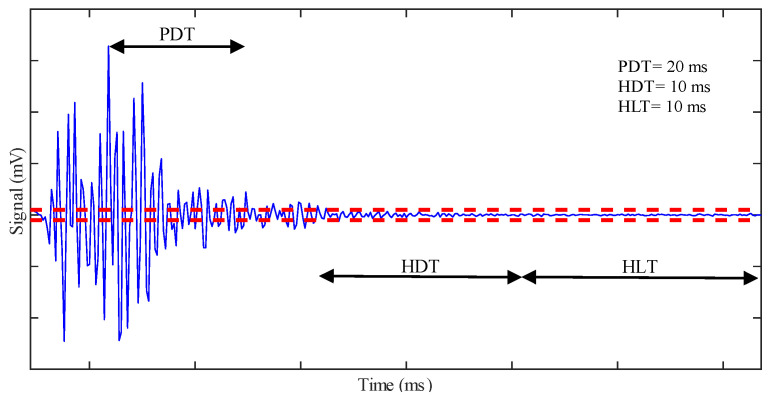
Peak definition time (PDT), hit definition time (HDT), and hit lockout time (HLT). They discriminate one hit from another.

**Figure 5 sensors-20-07272-f005:**
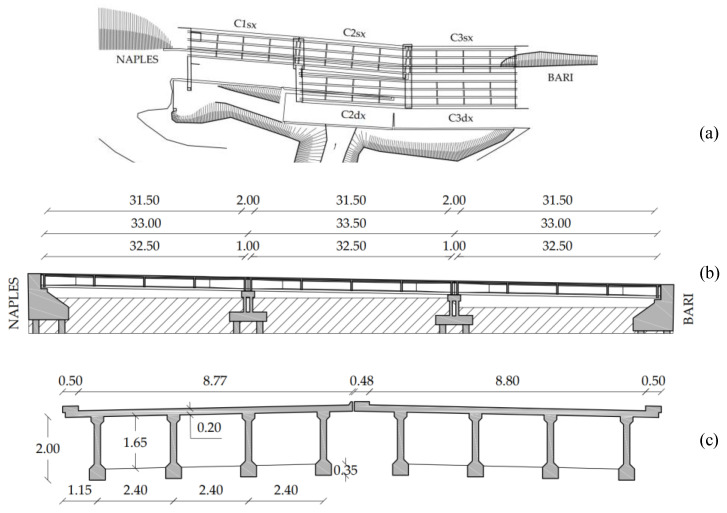
(**a**) Top view; (**b**) lateral view; and (**c**) cross section of the Alveo Vecchio viaduct (Italy).

**Figure 6 sensors-20-07272-f006:**
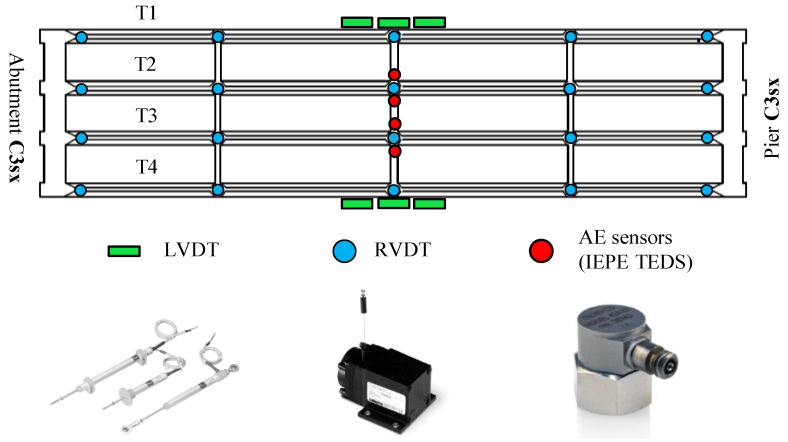
Monitoring system layout (only sensors relevant in our analysis).

**Figure 7 sensors-20-07272-f007:**
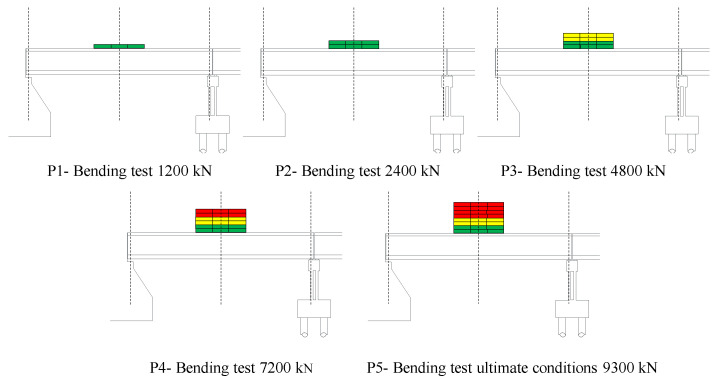
Load-test protocol: five loading phases with an increasing number of ballast weights.

**Figure 8 sensors-20-07272-f008:**
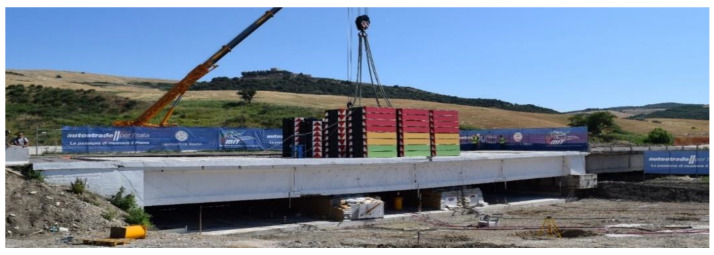
The Alveo Vecchio viaduct during the loading phase 5, loaded with 93 weights—9300 kN.

**Figure 9 sensors-20-07272-f009:**
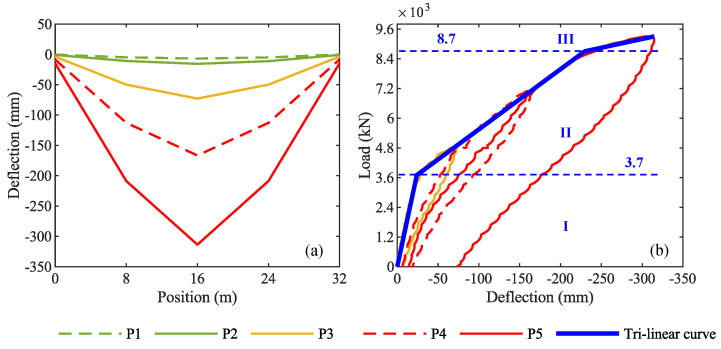
(**a**) Vertical displacements along girder T1 in the five loading phases; (**b**) load-deflection curve of girder T1 and its trilinear idealized flexural response.

**Figure 10 sensors-20-07272-f010:**
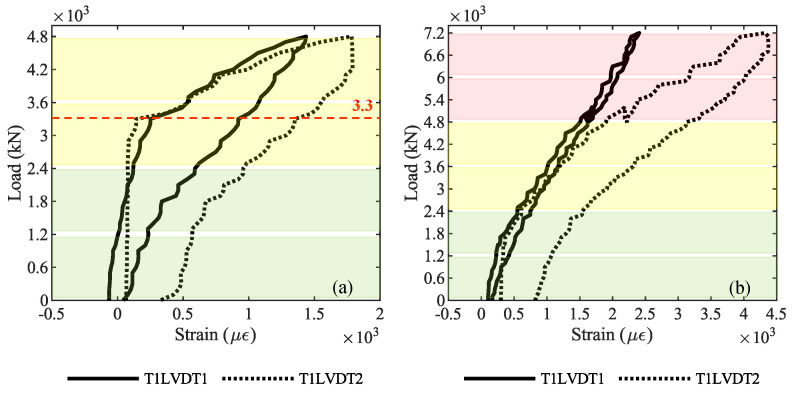
Longitudinal strain at the bottom of the middle cross section of girder T1: (**a**) loading phase P3; (**b**) loading phase P4. The red dashed line represents the first-crack load identified by crack-opening sensors.

**Figure 11 sensors-20-07272-f011:**
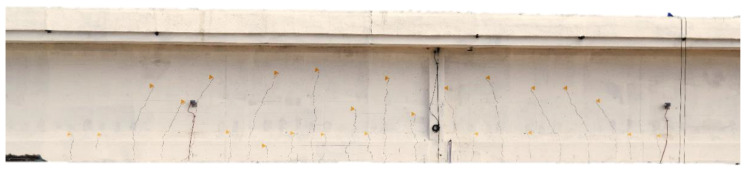
Visible cracks on girder T1 opened during the loading phase P3 4800 kN.

**Figure 12 sensors-20-07272-f012:**
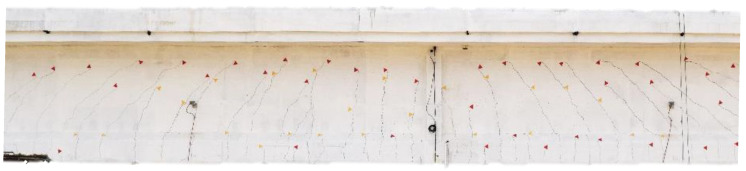
Visible cracks on girder T1 opened during the loading phase P4 7200 kN.

**Figure 13 sensors-20-07272-f013:**
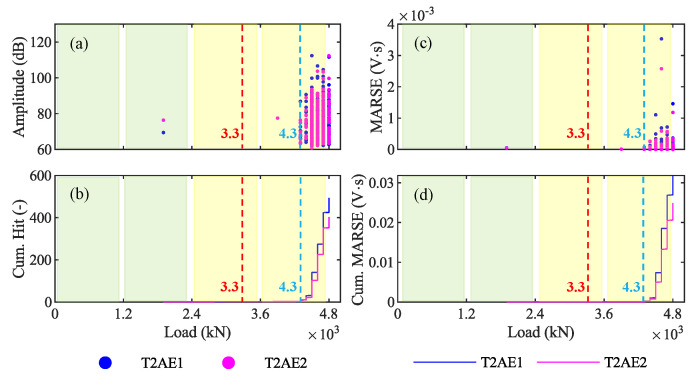
Results in the time-domain from the loading phase P3 4800 kN: (**a**) amplitude; (**b**) cumulative number of hits; (**c**) signal strength; and (**d**) cumulative signal strength.

**Figure 14 sensors-20-07272-f014:**
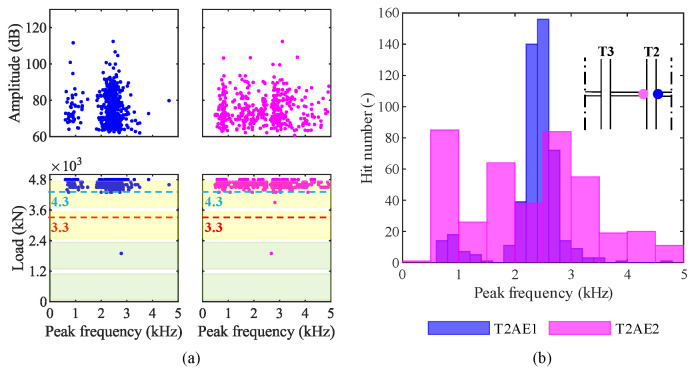
Results in the frequency-domain from the loading phase P3 4800 kN: (**a**) amplitude and load—peak-frequency; (**b**) peak frequency distribution among hits.

**Figure 15 sensors-20-07272-f015:**
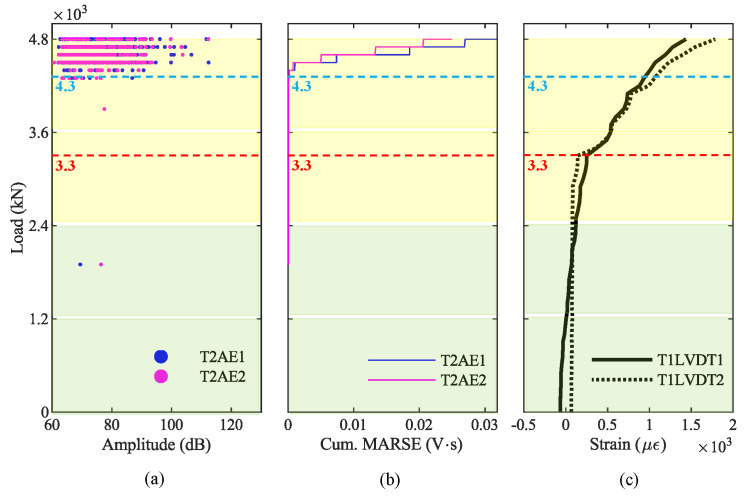
Results from AE and crack-opening sensors during phase P3: (**a**) amplitude from girder T2; (**b**) cumulative signal strength from girder T2; (**c**) longitudinal strain at the bottom of girder T1.

**Figure 16 sensors-20-07272-f016:**
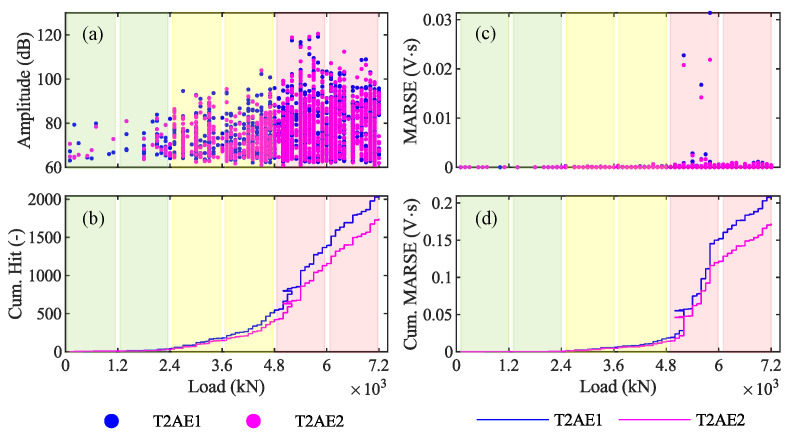
Results in the time-domain from the loading phase P4 7200 kN: (**a**) amplitude; (**b**) cumulative number of hits; (**c**) signal strength; and (**d**) cumulative signal strength.

**Figure 17 sensors-20-07272-f017:**
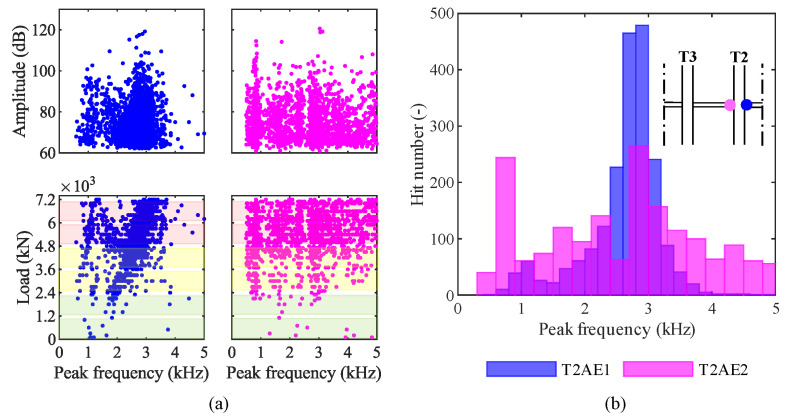
Results in the frequency-domain from the loading phase P4 7200 kN: (**a**) amplitude and load—peak-frequency; (**b**) peak frequency distribution among hits.

**Figure 18 sensors-20-07272-f018:**
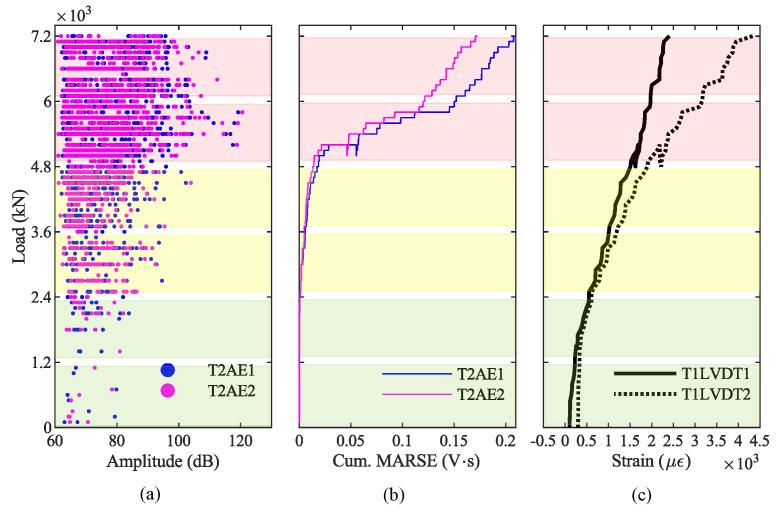
Results from AE and LVDT sensors during phase P4: (**a**) amplitude from girder T2; (**b**) cumulative signal strength from girder T2; (**c**) longitudinal strain at the bottom of girder T1.

**Figure 19 sensors-20-07272-f019:**
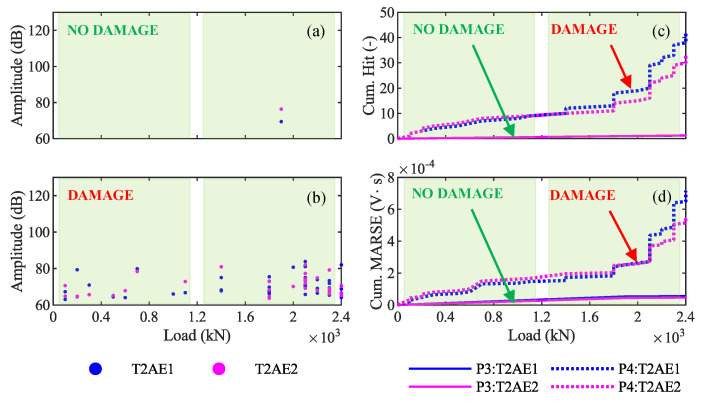
Differences in amplitude (**a**,**b**) and cumulative signal strength (**c**,**d**) between the AE acquired from the viaduct without pre-existent cracks (phase P3) and with pre-existent cracks (phase P4) for loads between 0 kN and 2400 kN.

**Figure 20 sensors-20-07272-f020:**
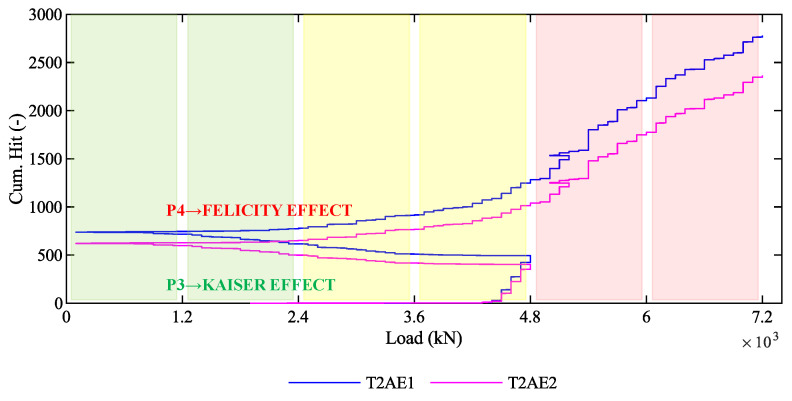
Kaiser and Felicity effects in the AE acquired during the loading phases P3 and P4.

**Table 1 sensors-20-07272-t001:** Materials properties of the Alveo Vecchio viaduct.

Slab Concrete Compressive Strength	Girders Concrete Compressive Strength	Post Tensioned Cables Residual Stress	Steel Yield Strength	Steel Ultimate Strength
f_cm_ (MPa)	f_cm_ (MPa)	σ_resm_ (MPa)	f_pym_ (MPa)	f_t1m_ (MPa)
31.9	41.5	952	1509	1618

**Table 2 sensors-20-07272-t002:** Technical features of sensors installed on the viaduct (only sensors relevant in our analysis).

Type	Full-Scale (FS)/Range	Accuracy	Sampling Frequency	Number
RVDT (deflection)	50–100 mm	1.5‰ FS	1 Hz	20
RVDT (deflection)	500 mm	5‰ FS	1 Hz	12
LVDT (crack-opening)	10 mm	1‰ FS	1 Hz	22
AE sensors	50 Hz–10 kHz	500 mV/g	10 kHz	4

**Table 3 sensors-20-07272-t003:** Threshold, sampling frequency, high-pass filter, and time-parameters PDT, HDT, and HLT.

Amplitude Threshold	Sampling Frequency	High-Pass Filter Frequency	Peak Definition Time (PDT)	Hit Definition Time (HDT)	Hit Lockout Time (HLT)
1 mV; 60 dB	10 kHz	500 Hz	20 ms	10 ms	10 ms

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
