# Peer review of "Structural Health Monitoring Based on Acoustic Emissions: Validation on a Prestressed Concrete Bridge Tested to Failure"

_sensors, 2020, doi:10.3390/s20247272_

Round 1
Reviewer 1 Report
The work is very interesting and suitable to be published in the present form. The quality of some figures must be improved.
The authors must discussed also in the conclusion the impact of moisture content on the efficiency of the proposed EA technique.
Author Response
Dear Reviewer, thank you for your comments. These are our responses:
- We updated a zip-file with all the graphs as vector graphics. Figures 5 has been printed on PDF from a CAD file at the maximum resolution. Figures 8, 11, and 12 have been took with a Reflex camera and they are in the original resolution.
- We reported the temperature and relative humidity recorded during the days of the load test in Section 4. We did not measure the changes in the moisture content in the concrete during our experiment and indeed, we could not find any experiment reported in the literature where the relation between the moisture and AE has been investigated. That would be a very interesting point to be addressed in future investigations and we truly appreciate that the Reviewer pointed that out.
Reviewer 2 Report
The work presented in the manuscript is interesting. But, the manuscript is poorly written and lags some important details. The author(s) can address my comments below:
- The abstract can be more concise.
- Too many 'however' is used.
- Please break the long sentences (e.g. Line 14, 17, 18, .....check the manuscript & modify similar sentences) and modify. Rewrite the sentence in Line 18: '..... carried on until their failure; therefore, there are still .....'.
- Line 25/26: '....an extensive monitoring system, consisting also of AE sensors....'. Remove unnecessary adjectives and adverbs.
- Line 49: acoustic emission
- Line 57: AE. Thoroughly check the manuscript to replace 'acoustic emission' with AE.
- Replace 'structural health monitoring' with SHM .....Line 61 onwards.
- Line 133: This contribution is organized as follow. Modify.
- Line 141: AE
- Fig. 1 is not required.
- The authors must include the details (with the photograph, with model no. and manufacturer's name.) of their AE signal monitoring system that controls the AE sensors.
- Include AE sensor description with type/model no. and manufacturer's name.
- Explain in detail: how the threshold and peak in the AE signals were identified?
- Conclusions: must be concise. Include the key research outcomes only.
- Finally, the manuscript should be thoroughly checked for grammar, typos, abbreviations.
Author Response
Dear Reviewer, thank you for your detailed comments. These are our responses:
- We revised the abstract. Now it is more concise and effective.
2-9. We truly appreciate your suggestions. We modified the manuscript according to them.
- We think that Figure 1 can help Readers who are not familiar with AE to understand the principle of AE monitoring. Anyway, we can remove it if the Reviewer doesn’t like it.
11-12. In Section 3.2, we added details on the model and the manufacturer’s name of all sensors we considered and their acquisition system.
- Details about the threshold are in Section 4, while details about the peak-frequency identification are in 2.2.
- We rewrote the conclusions according to your suggestions. Now they are concise and include only the main outcomes of the research.
- We had the manuscript checked by a translation agency as per the Reviewer’s request.
Reviewer 3 Report
The authors presented an application of the AE technique to a real-life case study. The test was monitored by an extensive monitoring system, consisting of AE sensors, whose most significant results we reported and discussed in this manuscript. It is a very interesting and practical research. The methodology was described comprehensively and the results are meaningful and valuable. Minor revision is required before it is published.
Please offer the full name for the term which is mentioned at the first time, but for the second time, only use the acronyms. For example, the Acoustic emission (AE) in line 141.
Some expressions in the manuscript are not clear. An English native speaker is suggested to carefully proofread it again. Normally, first-person pronouns are rarely used in scientific papers. The authors are requested to consider modifying some expressions to reduce the use of first-person pronouns.
The serial number of the figures should be consistent with the sequence of the figures mentioned in the text. Please modify the section 4.2.
Please provide more explanation about the why two clusters were recognized in the data from sensor T2AE1 in Figure 14 a.
Please provide more description on Figures 16 to 18. Otherwise, it is not necessary to show them.
In the conclusion section, the first paragraph is suggested to be deleted. It is not a real conclusion.
Author Response
Dear Reviewer, thank you for your detailed comments. These are our responses:
- Thank you for the suggestion: now there are acronyms wherever it is possible.
- We had the manuscript checked by a translation agency as per the Reviewer’s request.
- We deliberately used the first person and the active form instead of using the passive form extensively. Our personal opinion is that in modern scientific communication, the active form is advisable in most cases because it is more direct, clearer, and simpler to understand.
- Thank you for the suggestion: we fixed this error.
- We expanded the discussion of the results in the frequency domain in Section 5.1.
- Now Figures 16, 17, and 18 are discussed extensively in Section 5.2.
- We rewrote the conclusions according to your suggestions. Now they are concise and include the main outcomes of the research.
Round 2
Reviewer 2 Report
The manuscript can be accepted for publication